# Market Misreaction? Evidence from Cross-Border Acquisitions

**CNV Krishnan * and Jialun Wu**

Weatherhead School of Management, Case Western Reserve University, Cleveland, OH 44106, USA; jxw1253@case.edu
* Correspondence: cnk2@case.edu

**Abstract:** Our goal in this paper is to answer this research question: Do investors understand the longer-term value-implications of cross border mergers and acquisitions, as at the time of their announcements? We examine acquirers' operating efficiencies around and after cross-border acquisitions and relate this to the announcement-period stock-market reaction. Using a dataset of cross-border mergers and acquisitions (M&A) entailing U.S. acquirers over the period 1990–2013, and using a bootstrapped-DEA (Data Envelopment Analysis) model because any one indicator may not reflect the whole performance of the merger, we find that the operating efficiency of the acquirers decreases around the acquisition, and up to three years after. However, we document evidence of stock market mis-reaction at announcement: the announcement-period acquirer abnormal stock-price return is not significantly associated with acquirer's operating efficiency post-acquisition. Therefore, investors should be careful interpreting the announcement-period stock-price reaction in cross-border mergers and acquisitions as indicative of merger efficiency gains.

**Keywords:** cross-border M&A; merger efficiency; operating efficiency; abnormal returns





## 1. Introduction

Since 1985, more than 325,000 merger and acquisition (M&A) transactions have been announced, aggregating to more than USD 34,900 billion. With the rise of globalization, cross-border M&As have become increasingly prevalent. Cross-border M&As are arguably more complicated, involving more uncertainty around, for example, exchange rates and country features. It is important to assess whether cross-border M&As enhance operating efficiency for acquirers in the face of a recent trend towards de-globalization and trade frictions.

Our goal in this paper is to answer this research question: Do investors understand the longer-term value-implications of cross border mergers and acquisitions, as at the time of their announcements? We examine acquirers' operating efficiencies around and after cross-border acquisitions, and relate this to the announcement-period stock-market reaction.

We examine 822 cross-border mergers involving U.S. acquirers from the 1990 to 2013. Emerging markets may have weak contracting institutions, and it may be difficult for them to write enforceable contracts (Dyck and Zingales 2004), but acquirers from developed markets could bring better institutional mechanisms to targets (Chari et al. 2009). Hence, the challenges that acquirers face may be offset by the advantages that the cross-border M&As could bring (e.g., Hofstede 1980; Aybar and Ficici 2009). That is why we chose U.S. acquirers and examine their announcement period stock price reactions.

We analyze the three-year operating performance of the acquirers from and including the year of the M&A, and we compare this to three-year performance before the acquisition. We employ a Bootstrap-DEA (data envelopment analysis) model (Cooper et al. 2000) instead of parametric tests. A DEA model is effective in dealing with complex production process entailed in cross-border mergers because it is appropriate for a setting with multiple inputs and multiple outputs.

We find that the acquirer operating efficiency was, on average, significantly higher two and three years before the acquisition announcement, as compared to the year of the announcement. Three years after the acquisition, the operating efficiency decreases significantly further than the transaction year. However, we find that the announcement-period acquirer cumulative abnormal return (CAR) has no significant relationship with post-merger efficiency. Indeed, acquirers that experienced negative CAR experienced a significant decrease in operating efficiency pre-acquisition, while the acquirers with positive CAR did not. Therefore, investors should be careful interpreting the announcement-period stock-price reaction in cross-border mergers and acquisitions as indicative of merger efficiency gains post-acquisition.

The next section provides a relevant literature review, and Section 3 describes our data and the variables we use. The main analyses of returns and performance, along with results discussions and robustness checks, are reported in Section 4. Section 5 concludes.

## 2. Literature Review

Extant literature has discussed M&A benefits for acquirers that may depend on merger waves (Martynova and Renneboog 2008), which include developing access to foreign customer base (Luo and Tung 2007), economies of scale and scope, lower transaction cost (because some resources and skills may not available for the acquirers in the domestic market), and improved capacity utilization. Acquirers could enhance market power and create enterprise value (Gugler et al. 2003). Hence, cross-border M&As may increase the efficiency of business operations, develop new opportunities for growth (Bertrand and Betschinger 2012), and offset any competitive disadvantages (Luo and Tung 2007). Cross-border acquisitions could benefit acquirers more than domestic acquisitions (Chari et al. 2009), because, for example, acquirers may generate valuation gains when the acquirer's country has stronger investor protections than the target's country (Bris et al. 2003). Cross-border M&As may also help acquirers diversify operating risk (Severn 1974). On the other hand, cross-border acquirers may face challenges that may offset the benefits (Aybar and Ficici 2009). For instance, the firm may have to face political risk, exchange rate risk, and new cultural environments. Cultural differences between the acquirers and targets could have a negative impact on the acquirer's long-term abnormal returns (Hofstede 1980). A target's country's policy uncertainty may make the acquirer become cautious when making cross-border merger decisions (Pástor and Veronesi 2012, 2013).

In short, whether cross-border acquisitions improve an acquirer's operating efficiency is ambiguous.

We examine the announcement period abnormal stock returns of the acquirers around M&A announcements. Announcement-period acquirer-stock prices likely react to the possibility of, or the lack of, expected synergies between the organizations, integration issues and restructuring plans (Angwin 2001). Several papers have analyzed abnormal returns around M&A announcement or effective dates (e.g., Hofstede 1980; Masulis et al. 2007; Aybar and Ficici 2009; Francis and Martin 2010; Harford et al. 2012; Ahern et al. 2012; McNichols and Stubben 2015), internal controls (e.g., Chen et al. 2016; Albuquerque et al. 2018; Surbhi and Vij 2018), or acquirer's operating performance (e.g., Francis and Martin 2010). However, it is likely that stock returns around announcement reflect short-term market sentiment about the deal, but may not fully reflect competitive advantages, or the lack thereof, in the longer run from cross-border M&As. Investors may also be nervous about the successful completion of the M&A (Angwin 2004), and there may be information asymmetry between insiders and investors (Graffin et al. 2011; Zhang 2008), leading to misreaction. Indeed, Krishnan and Yakimenko (2021), also document misreaction as at the time of announcements to bank and non-bank domestic mergers, based on leverage.

We employ a Bootstrap-DEA (Data Envelopment Analysis) model (Cooper et al. 2000) instead of parametric tests, to analyze the three-year operating performance of the acquirers from and including the year of the M&A and compare to performance before the acquisition. A DEA model is effective in dealing with complex production process entailed in cross-

border mergers because it is appropriate for a setting with multiple inputs and multiple outputs (Schaffnit et al. 1997; Hartman et al. 2001; Paradi and Zhu 2013). Further, with the DEA model, we could analyze each DMU (Decision Making Unit) individually and could identify inefficient DMUs according to the benchmarks (Aggelopoulos and Georgopoulos 2017; Repková 2014). Further, no preconceived structure need be imposed on the data when determining the efficient frontier (Avkiran 1999). However, a conventional DEA model has some limitations related to the precision of the estimation of the efficient frontier, so we use the bootstrap procedure (Aggelopoulos and Georgopoulos 2017). For selecting the input and output variables, we refer to the previous literature that use DEA models to measure operating efficiency (Ropero et al. 2019; Halkos and Tzeremes 2013; Asmild et al. 2009; Sherman and Gold 1985). Some of this literature use the DEA model for specific industries; for instance, Liu (2012) analyze steel industry's merger efficiency, Halkos and Tzeremes (2013) analyze the operating efficiency from bank M&As, and Ropero et al. (2019) analyze the operating efficiency in container ports by adding specific variables as input and output variables. In the paper, we analyze cross-border merger efficiency of all U.S. enterprises instead of a specific industry or industries, so we select more "general" variables. We choose total assets, total cost, and total operating expense (including management cost, operating cost, and financial cost) as the input variables.

Total assets are the resources used for generating profits. Total cost as well as operating expense, consisting of management cost, operating cost, and financial cost, are measures of cost effectiveness. These are our input variables. Net income improvements may be viewed as a positive outcome of an acquisition; prime operating revenues reflects the productivity of the acquirer; and total debt may also be viewed as an output of the acquisition, especially after levered transactions. These are our output variables.

## 3. Data, Variables and Methodology

Our data come from Refinitiv's SDC Platinum Mergers and acquisitions database, which contains 1512 cross-border acquisitions, over a 23-year period from 1990 through 2013. From the original dataset, we exclude all observations that do not have information on all details of the transactions that we need. Our final dataset contains 822 cross-border acquisitions.

### 3.1. Methodology

We employ a bootstrap-DEA (data envelopment analysis) model instead of parametric tests to analyze the three-year operating performance of the acquirers from and including the year of the M&A and compare to performance before the acquisition. We then examine the difference in operating efficiency between different years and the reference year, the year in which the acquisition happened. We also compute the announcement period cumulative abnormal returns (CARs) over different windows for the acquirers and relate to operating efficiency changes. We check these associations using multivariate tests to examine whether the acquirers that have positive abnormal return tend to have positive post-merger efficiency. We control acquirers' characteristics and target nations' characteristics. Finally, we perform several robustness checks that include PSM Propensity Score Match) method, using which we can avoid any uncontrolled-for factors, we examine alternative announcement period abnormal returns (computed over and above different benchmarks and over different periods around the announcement date), and we examine results on sub-samples with positive post-merger operating efficiency and negative post-merger operating efficiency. These are all detailed in the next sections.

### 3.2. Merger Efficiency, Independent Variables and Control Variables

Merger efficiency is defined as the operating efficiency of the acquirers in the three years following the acquisition compared to the operating efficiency of the acquirer in the year that the acquisition happened. We employ the bootstrap-DEA model to measure operating efficiency, in which, as mentioned above, total assets, total cost, and total operating

expense (including management expense, sales expense, and financial expense) as the input indices, and net income, prime operating revenues, and total debt are the output indices. The DEA model is widely used to assess the efficiency (Charnes et al. 1978; Guijarro et al. 2020), and involves estimating the efficient frontier, and then comparing decision making units (DMUs) to efficient DMUs.

In general, one DMU would have a set of inputs $X = (x_1, x_2 \ldots x_s)$ and produce a set of outputs $Y = (y_1, y_2 \ldots y_m)$, and under the resource conservation assumption, the efficiency score $\theta$ is between 0 and 1 ($0 \le \theta \le 1$). Therefore, for the efficiency score of the $j_0$ DMU, we could obtain the following equation:

$$\max h_0 = \frac{\sum_{r=1}^{s} u_r * y_{rj_0}}{\sum_{i=1}^{m} v_i * x_{ij_0}}$$

subject to:

$$\frac{\sum_{r=1}^{s} u_r * y_{rj}}{\sum_{i=1}^{m} v_i * x_{ij}} \le 1, \ j = 1, 2, \ldots n$$

$$v = (v_1, \ v_2, \ldots, v_m)^T \ge 0$$

$$u = (u_1, u_2, \ldots, u_s)^T \ge 0$$

Here, $u_r$ and $v_i$ are the variable weights determined by the solution of the problem. The efficiency of the $j_0$ DMU is to be analyzed relative to other DMUs.

We non-dimensionalize these variables. In this model, different variables have different dimensions, and some variables could even be negative, such as the profit; this would affect the accuracy of the model. The dimensionless quantity would help us obtain a quantitative gauging of features without loss of any information. Because of the advantages, previous papers have employed it in the DEA model (Arana-Jiménez et al. 2020; Tofallis 2014; Asmild and Pastor 2010; Carlos and Román 2001; Lovell and Pastor 1995). To remove dimensionality, we rank the $p$-index and select the maximum ($a_p$) and the minimum ($b_p$) in the adjusted index:

$$Z_{jp}^* = 60 + \frac{Z_{jp} - b_p}{a_p - b_p} * 40, \qquad Z_{jp}^* \in [60, \ 100]$$

We select six variables to measure the country governance of target: voice and accountability (VA), political stability and absence of violence/terrorism (PV), government effectiveness (GE), regulatory quality (RQ), rule and law (RL) and control and corruption (CC) (see Kaufmann et al. 2010). Kaufmann et al. (2010) believe this to be a useful way of organizing and summarizing a very large and disparate set of individual perception-based indicators of governance that have become available since the late 1990s. The data are taken from http://info.worldbank.org/governance/wgi/ accessed on 9 August 2018. To study the effect of religiosity, we use an indictor variable: Rates of Adherence per 1000 Population (County) (the religiosity of the county the acquirer is located in).

As control variables, we use the target nation's characteristics and the acquirer's characteristics, as these variables could affect merger outcomes. The target nation's characteristics include GDP growth, external investment, inflation rate, interest rate, and tax rate. GDP reflects target nation's economic vitality: international investors tend to invest in the country where the economy has grown fast (Choi 2003). External investment may reflect the appeal for foreign investment in the target nation—the target nation's investment environment. A low and stable inflation rate could reduce the macroeconomic risks in the host country's market and may make the country a more attractive destination (Asongu et al. 2018). Previous literature has indicated that interest rate may play a pivotal role in attracting the foreign investment (Asiamah et al. 2019; Saini and Singhania 2017; Reenu and Sharma 2015). Countries with lower tax rates may also be more attractive for foreign investors.

Acquirers' characteristics include size, leverage, profit, cost, expense, and income. Size is natural log of total asset of the acquirers taken from annual financial reports; leverage is the ratio of the total debt to total assets; profit is the ratio of net income to prime operating revenue; cost is the total cost divided by the prime operating revenue; expense is the sum of the management expense, financial expense, and sales expense divided by the prime operating revenue; and income is prime operating revenue divided by total asset. All these variables reflect the general operating and financial condition of the acquirer. Moreover, the value of the transaction could also affect the operating efficiency in the following year—if the size of the transaction is small, the acquirers would not be affected as much. All these variables are taken from annual Compustat database. Descriptions of variables are in Appendix A.

### 3.3. Descriptive Statistics

Table 1 shows the number of cross-border acquisitions for different periods. The biggest group is in the period 1996–2001 as it is in the fifth M&A wave. The table indicates that after 2008, the pace of U.S. cross-border acquisitions slowed down—the economic crisis of 2008 affected cross-border acquisition activities. Devaluation of the U.S. dollar increased transaction values and decreased numbers.

**Table 1.** Sample Descriptive Statistics.

| Period | Number | Transaction Value ($ mil) |
|---|---|---|
| 1990–1995 | 153 | 15,697 |
| 1996–2001 | 321 | 85,152 |
| 2002–2007 | 202 | 104,661 |
| 2008–2013 | 146 | 99,100 |
| Total | 822 | 304,610 |

Table 1 shows the number of observation and total value of cross-border mergers and acquisitions with U.S. acquirer for different periods. The data are taken from Refinitiv's SDC Platinum Mergers and acquisitions database spanning the period 1990–2013.

Table 2 shows the descriptive statistics of the bootstrap-DEA model indices. We analyze the acquirer's operating efficiency around the acquisition year. From the table, we find that the first quartile of net income is zero, indicating, on average, acquirers may be seeking cross-border acquisitions to improve their operating efficiency. Additionally, from Table 2, we notice the standard deviation of the indices—total asset, total cost, total operating expense, net income, prime operating revenue, and total debt are all high—hence, employing the bootstrap method is necessary to reduce the impact of extremes.

**Table 2.** Descriptive statistics of bootstrap-DEA model Indices.

| | N | Mean | SD | Median |
|---|---|---|---|---|
| Total Asset (USD mil) | 5486 | 74,680 | 281,640 | 4492 |
| Total Cost (USD mil) | 5486 | 6572 | 16,966 | 962 |
| Total Operating Expense (USD mil) | 5486 | 3261 | 12,459 | 103 |
| Net Income (USD mil) | 5486 | 751 | 2714 | 71 |
| Prime Operating Revenue (USD mil) | 5486 | 14,442 | 28,461 | 3248 |
| Total Debt (USD mil) | 5486 | 64,921 | 258,666 | 14,088 |

Table 2 shows the descriptive statistics of input and output indices used in the bootstrap-DEA model used to get the operating efficiency: the number of observations (N), mean, standard deviation (SD), and median. These descriptive statistics describe original values instead of non-dimensionalized values. The data come from the Compustat database.

Table 3 shows the descriptive statistics of the control variables in the difference-in-difference (DID) model to model the treatment effect (whether acquisitions with positive announcement period abnormal returns entail higher post-merger efficiency) by estimating the difference between the outcome measures at two time points for both the treatment observations and controls. The reason why we use this model is that we need to compare the difference of the merger efficiency between the acquisition with positive announcement period cumulative abnormal return (CAR) of the acquirer and those with negative CAR, where CAR is defined as return of the acquirer's stock minus beta times S&P return, where beta is estimated over the year prior to the announcement.

**Table 3.** Descriptive statistics of explanatory variables of the DID model.

|  | N | Mean | SD | Median |
| --- | --- | --- | --- | --- |
| Size | 5486 | 8.37 | 2.46 | 8.43 |
| Leverage | 5486 | 2.32 | 2.45 | 1.75 |
| Profit | 5486 | −11.39 | 456.63 | 0.04 |
| Income | 5486 | 0.85 | 0.67 | 0.72 |
| Cost | 5486 | 11.08 | 449.94 | 0.59 |
| Expense | 5486 | 0.59 | 13.50 | 0.03 |

Table 3 shows the descriptive statistics of explanatory variables of the difference-in-difference (DID) model. These are six acquirer's indicators: the size, leverage, profitability, income level, cost level and expense level: the number of observations (N), mean, standard deviation (SD), and median. The data come from the Compustat database. All variables are described in Appendix A.

The premise of the DID model is that constant difference in outcome exists in the absence of the treatment, which we would need the common trend assumption test, for which we test. From the table, we note that the standard deviations of profit and cost are high (the standard deviation of the profit is 456.63, while the mean of the profit is −11.39. the standard deviation of the cost is 449.94, whereas the mean of the cost is 11.08). Thus, there is a large variability in profitability and cost management ability among acquirers. To remove the effect of outliers, before conducting the DID analysis, we winsorize these two variables—profit and cost—at 1% level, and the standard deviation drops dramatically (the standard deviation for profit is now 0.67, and that for the cost drops to 0.45). Table 4 shows the descriptive statistics of the variables of the regression model. There are no major concerns with outliers.

**Table 4.** Descriptive statistics of variables in regression model.

|  | N | Mean | SD | Median |
| --- | --- | --- | --- | --- |
| CC | 5486 | 0.64 | 0.22 | 0.71 |
| GE | 5486 | 0.85 | 0.14 | 0.89 |
| RL | 5486 | 0.79 | 0.14 | 0.75 |
| RQ | 5486 | 0.87 | 0.14 | 0.92 |
| PV | 5486 | 0.85 | 0.14 | 0.88 |
| VA | 5486 | 0.80 | 0.13 | 0.82 |
| TOTRATE | 5486 | 480.45 | 127.27 | 440.75 |
| VOT | 5486 | 0.09 | 0.24 | 0.01 |
| INV | 5486 | $2.98 \times 10^{-6}$ | $7.15 \times 10^{-6}$ | $1.10 \times 10^{-6}$ |
| GDPG | 5486 | 2.86 | 2.68 | 3.09 |
| IR | 5486 | 3.37 | 6.94 | 2.11 |
| T | 5486 | 16.59 | 7.15 | 14.00 |
| R | 5486 | 6.46 | 8.90 | 5.81 |

Table 4 reports the descriptive statistics of variables in our regression model. The country-level governance variables are: control and corruption (CC), government effectiveness (GE), rule and law (RL), regulatory quality (RQ), political stability and absence of

violence/terrorism (PV) and voice and accountability (VA). We use the Rates of Adherence per 1000 Population (TOTRATE) to measure the religiosity of the county that the acquirers located in. We use the country's characteristics as the control variables: GDP growth (GDPG), inflation rate (IR), interest rate (R), tax rate (T) and external investment (INV), which is total external investments divided by GDP of target nation. The value of the transaction (VOT) and the acquirer's characteristics are also selected as control variables. The number of observations (N), mean, standard deviation (SD), and median are reported. The data come from http://info.worldbank.org/governance/wgi/, accessed on 9 August 2018. All variables are described in Appendix A.

## 4. Post-Merger Efficiency

### 4.1. Operating Efficiency

Following (Pawlowska 2003; Rahman and Lambkin 2016; Wang and Zhang 2018), we employ a bootstrap-DEA model with input and output indexes and remove dimensionality, as described above. To avoid any bias caused by extreme values because of small sample size, we introduce bootstrap as follows:

First, obtain the initial efficiency value by DEA model $\theta = \{\theta_k | k = 1, 2 \dots n\}$; then, use bootstrapping by repeating and putting back samples to calculate the efficiency value of the sample size of n: $\theta_{1b}, \theta_{2b}, \theta_{3b}, \dots \theta_{nb}$, where b is the number of the iteration. Because the efficiency that we obtain is not consistent with the general density curve, we need to smooth the efficiency that we get.

$$\overline{\theta_b} = \left\{ \overline{\theta_{bk}} \middle| k = 1, 2, \dots n \right\} \quad b = 1, 2, \dots n$$

We use $\overline{\theta_{bk}}$ to adjust the input index.

$$X_{bk} = \left( \frac{\theta_k}{\theta_{bk}} \right) * X_k$$

We then use the adjusted input index and output to obtain the adjusted $\theta_b^*$.

$$\theta_b^* = (\theta_{bk}^* | k = 1, 2, 3 \dots n)$$

Finally, we repeat the above steps *n* times and calculate the estimation of the bias and the adjusted efficiency.

$$\widehat{bias_k} = \frac{\sum_{b=1}^{n} \theta_{bk}^*}{n} - \theta_k$$

$$\theta_k^{adj} = \theta_k - \widehat{bias_k}$$

The above process is used over all data to obtain the operating efficiency for the acquirers for 7 years around the year of acquisition announcement $[-3, +3]$.

### 4.2. Univariate Tests

Table 5, panel A, examines the average efficiency value for acquirers and reports the *Mann–Whitney U* test statistics of the difference in operating efficiency between different years and the reference year, the year in which the acquisition happened. Panel B (C) reports the results for acquisitions with positive (negative) CAR. From panel A, we observe that the mean efficiency value decreases as time goes by. In fact, the operating efficiencies in year $-2$ (two years before the acquisition) and year $-3$ (three years before the acquisition) are significantly higher than the efficiency in the year 0 (the year the acquisition happened). Additionally, the operating efficiency in the year $+3$ (the third year after the acquisition) is significantly lower than the efficiency in the year 0.

Panels B and C also show that the operating efficiency in year $+3$ is significantly lower than the efficiency in the year 0. However, we notice that acquisitions with negative announcement period CARs had significantly higher efficiency, on average, in year $-2$ and

year $-3$ than in the acquisition year; that is, efficiency decreased significantly through to the announcement year. On the other hand, acquisitions with positive announcement period CARs have not had significantly higher efficiency previously in the year $-2$ and year $-3$ as compared to the acquisition year; that is, efficiency has not decreased significantly through to the announcement year. Hence, announcement period CAR may reflect operating efficiency changes in the recent *past*, not the expected one.

Table 5 is the analysis of the operating efficiency around the acquisition year. Panel A shows the average of the operating efficiency ($\hat{\theta}_k$) in the seven years around the M&A (for instance, the third year after the acquisition is showed as +3, and third year before is $-3$). Panel A also reports the Mann–Whitney U-test of the difference of the operating efficiency of another year and the announcement year. Panel B and panel C report the average of the operating efficiency and the Mann–Whitney test result for different type acquisitions—those with positive cumulative abnormal return and those with negative cumulative abnormal return, respectively.

**Table 5.** Analysis of the operating efficiency around the acquisition year.

| Panel A | | |
|---|---|---|
| **Year** | $\widehat{\theta_k}$ | **Mann–Whitney U-Test** |
| +3 | 0.999142 | 0.000 *** |
| +2 | 0.999349 | 0.095 |
| +1 | 0.999379 | 0.290 |
| 0 | 0.999410 | |
| −1 | 0.999492 | 0.059 |
| −2 | 0.999517 | 0.007 *** |
| −3 | 0.999511 | 0.001 ** |
| **Panel B** | | |
| **Year** | $\widehat{\theta_k}$ | **Mann–Whitney U-Test** |
| +3 | 0.999242 | 0.000 *** |
| +2 | 0.999455 | 0.088 |
| +1 | 0.999493 | 0.284 |
| 0 | 0.999512 | |
| −1 | 0.999585 | 0.239 |
| −2 | 0.999595 | 0.107 |
| −3 | 0.999592 | 0.055 |
| **Panel C** | | |
| **Year** | $\widehat{\theta_k}$ | **Mann–Whitney U-Test** |
| +3 | 0.99903 | 0.000 *** |
| +2 | 0.999233 | 0.512 |
| +1 | 0.999251 | 0.643 |
| 0 | 0.999297 | |
| −1 | 0.999388 | 0.134 |
| −2 | 0.999432 | 0.030 ** |
| −3 | 0.999420 | 0.009 *** |

** significant at 5% level, and *** significant at 1% level.

### 4.3. Multivariate Tests

We use the DID method to examine whether the acquirers that have positive abnormal return tend to have positive post-merger efficiency.

$$\theta_k^{adj} = \beta_1 * Positive\ CAR + \beta_2 \times Announcement\ Period +$$
$$\beta_3 * Positive\ CAR \times Announcement\ Period + \tag{1}$$
$$\sum Year\ Dummy + \sum IND\ Dummy + Z_{i,t} + \varepsilon_{it}$$

If the announcement period abnormal return is positive, then the dummy variable *Positive CAR* = 1; otherwise, *Positive CAR* = 0. *Announcement Period* is a dummy variable such that if the acquisition happened in that year, *Announcement Period* = 1; otherwise, *Announcement Period* = 0. $Z_{i,t}$ is a set of control variables, including Acquirer's Size (*Size*), Leverage (*Lev*), Profit (*Pro*), Income (*Inc*), and Cost (*Cost*). To employ the DID model, the sample should meet the common trend assumption (requiring that, in the absence of the treatment, the difference between the "treatment group" and "control group" would be fixed over the time). Hence, we use Model (2) to test whether the sample meets the common trend assumption:

$$\theta_k^{adj} = \alpha + \sum_{1}^{t} \beta_t \times Positive\ CAR \times Year\ Before_t + Z_{it} + u_i + v_t + \varepsilon_{i,t} \qquad (2)$$

Here, *Year Before*$_t$ is the dummy variable for the years before the acquisition; in particular, 3 years before acquisition and 2 years before acquisition. We run the regression model for the acquirers after the acquisition as below:

$$\theta_k^{adj} = \alpha_0 + \sum_{i=1}^{6} \beta_i \times CG_i + \beta_7 * Val + \beta_8 \times REL + Z_{i,t} + \sum Year\ Dummy + \varepsilon_i \qquad (3)$$

where $\theta_k^{adj}$ is the operating efficiency after the acquisition, and $CG_i$ is a vector of country governance, which consists of six variables: voice and accountability (VA), political stability and absence of violence/terrorism (PV), government effectiveness (GE), regulatory quality (RQ), rule and law (RL), and control and corruption (CC). *Val* is the value of the transaction, computed as the ratio of the value of transaction and the total asset to measure the value of the transaction. *REL* is used to measure the degree of religiosity in the county where the firm located and is calculated as the number religious adherents in the county (as reported by *Association of Religion Data Archives*) to the total population in the county (see Hilary and Hui 2009).

Control variables consist of acquirer's characteristics and target nation's characteristics. The acquirer's characteristics include Acquirer's Size (Size), Leverage (Lev), Profit (Pro), Income (Inc), and Cost (Cost), and the target nation's characteristics include GDP Growth (GDPG), Inflation Rate (IR), Interest rate (R), Tax rate (T), and External Investment (INV).

From Table 6, panel A, we find that the *p*-values of the *Positive CAR* $\times$ *Year*$_{-3}$ and *Positive CAR* $\times$ *Year*$_{-2}$ are not significant at the 5% level; therefore, whether the CAR is negative or positive has no strongly significant associations with operating efficiency before the acquisition. Hence, we conclude that the sample meets the Common Trend Assumption, and we could employ the DID model to analyze whether acquirers with different CAR entail different merger efficiency.

From Panel B, we notice that the variable *Positive CAR* is significant (*p*-value is about 0.015); however, the *p*-values of the variable *Announcement Period* and *Positive CAR* $\times$ *Announcement Period* reveal that these two variables are not significant (*p*-value for *Positive CAR* $\times$ *Announcement Period* is 0.824). This term *Positive CAR* $\times$ *Announcement Period* plays a pivotal role in determining whether CAR predicts/affects post-merger efficiency. Hence, announcement period reaction is not significantly associated with post-merger efficiency.[1]

Table 6 shows regression coefficients, *p*-values in parentheses, and the adjusted $R^2$. The premise that we could employ the difference-in-difference (DID) model is that the sample meets the common trend assumption. Hence, we report the result of Model (2) in panel A and the results of Model (1) in panel B. In Model (2), we only analyze the sample of observation before the acquisition; hence, the number of the observation is 2260 instead of the 5486.

**Table 6.** CAR and Merger Efficiency: DID Analysis.

| | Panel A | | | |
| --- | --- | --- | --- | --- |
| | Positive CAR × Year$_{-3}$ | Positive CAR × Year$_{-2}$ | Control Variables | Adj-R$^2$ |
| Coefficient | 0.00002 | 0.00002 | Included | 0.08 |
| *p*-Value | 0.359 | 0.364 | | |
| | Panel B | | | |
| | Positive CAR | Announcement Period | Positive CAR × Announcement Period | Control Variables | Adj-R$^2$ |
| Coefficient | 0.00016 | −0.00013 | −6.13 × 10$^{-6}$ | Included | 0.14 |
| *p*-Value | 0.015 ** | 0.731 | 0.824 | | |

** significant at 5% level.

From Table 7, we notice that country governance has only a little influence on the acquirer's operating efficiency after the acquisition, because only one variable, *Rule and Law*, of the six variables that reflect the country governance, affects operating efficiency at 1% significance level. Although *Rates of Adherence* has negative impact on the post-merger operating efficiency, the impact is not significant *(p*-value is 0.559). Hence, religiosity is not significantly associated with acquirer's post-acquisition efficiency. However, all acquirer's characteristics affect the post-acquisition operating efficiency significantly at the 1% level. *Leverage*, *Profit level*, *Cost*, and *Expense* all affect the operating efficiency negatively. Larger acquirers tend to have better operating efficiency after the acquisition. Most characteristics are associated with post-acquisition operating efficiency in expected ways. Perhaps the surprising result here is the negative association of *Profit level* with post-merger efficiency. One explanation could be hubris (Roll 1986): acquirers with high current profitability may pay higher acquisition premium, which would lower post-merger efficiency.

Table 7 shows regression coefficients, *p*-values in the parenthesis, and R$^2$ of Model (3). Explanatory variables consist of: country governance, religiosity, target nation's characteristics, acquirer's characteristics and value of transaction. Because we only analyze the sample after the acquisition, N is 2129 instead of 5486. All variables are defined in Appendix A.

**Table 7.** Determination of merger efficiency: multivariate analysis.

| Variable Name | Coefficient | *p*-Value |
| --- | --- | --- |
| Control and Corruption | −0.00021 | 0.523 |
| Government Effectiveness | 0.00025 | 0.667 |
| Rule and Law | −0.00111 | 0.003 *** |
| Regulatory Quality | −0.00086 | 0.153 |
| Political Stability and Absence of Violence/Terrorism | 0.00054 | 0.212 |
| Voice and Accountability | 0.00074 | 0.123 |
| Rates of Adherence per 1000 Population | −1.52 × 10$^{-7}$ | 0.599 |
| Size | 2.26 × 10$^{-7}$ | 0.000 *** |
| Leverage | −1.82 × 10$^{-7}$ | 0.000 *** |
| Profit | −1.14 × 10$^{-7}$ | 0.000 *** |
| Cost | −2.99 × 10$^{-7}$ | 0.000 *** |
| Expense | −2.49 × 10$^{-7}$ | 0.000 *** |
| Income | 2.10 × 10$^{-7}$ | 0.000 *** |
| Value of Transaction | 0.00072 | 0.000 *** |
| External Investment | −24.06624 | 0.000 *** |
| GDP Growth | 0.00004 | 0.008 *** |
| Inflation Rate | −6.02 × 10$^{-6}$ | 0.297 |
| Tax Rate | 0.00002 | 0.016 ** |
| Interest Rate | −1.90 × 10$^{-6}$ | 0.745 |

** significant at 5% level, and *** significant at 1% level.

The value of the transaction or the relative size—ratio of the value of transaction to the total assets of the acquirer—also matters. For the acquirers, there is evidence that the relative size of the target to the bidder matters (Asquith et al. 1983). Integration cost may be determined by the relative size of the merger transaction (Malmendier et al. 2018; Ahern 2012). For acquirers, larger acquisitions may have greater synergy potential (Kitching 1967). Larger integration benefits because of relative size of the acquisition may improve post acquisition operating efficiency. Overpayment potential may also be lower in acquisitions of big targets (Alexandridis et al. 2013). Indeed, we see from Table 7 that relative size is significantly positively associated with merger efficiency.

Among target nation's characteristics, *GDP growth* and *Tax rate* affect operating efficiency positively, as expected, but external investment (ratio of foreign investment to GDP) has a significant negative association with operating efficiency. This may be surprising. One explanation could be that, for countries that already have had higher external investment, their incentives for offering preferential treatment to new cross-border investments may be relatively lower, as compared to countries actively seeking such investments.

Note that all coefficients are close to zero. This is in line with extant literature: when they use efficiency as the dependent variable, the coefficients are close to zero (Liu 2012). This is because the efficiencies are low; most of them are between 0.9 to 1.0. Further, from Mann–Whitney statistics, the efficiency decreases significantly post-acquisition.

### 4.4. Results and Discussion

Our results that the competitive advantages acquirers may obtain from cross-border acquisitions may not offset the challenges such acquisitions bring and may be contrary to the conclusions in some of the extant literature, which show that cross-border acquisitions may entail positive merger efficiency. However, we have been careful to use the correct econometric methodology: a bootstrapped-DEA model, because in a merger, any one indicator may not reflect the whole performance of the merger, and this is the more appropriate model to use in these situations (Rahman and Lambkin 2016; Halkos and Tzeremes 2013; Liu 2012).

Religiosity and target nation's country governance features also have no significant associations with post-merger efficiency. However, acquirers' financial features—total assets, leverage, and profits—and target nation's macroeconomic features such as GDP growth, external investment level, and tax rate play more pivotal roles in being associated with operating efficiency in the years after the acquisition.

The main result is that the announcement period reaction is not significantly associated with post-merger efficiency. A couple of examples, discussed below, illustrate this finding.

On 20 December 2012, Flir Systems, which is a world leader in the design, manufacture, and marketing of sensor systems, announced that it acquired Lorex Technology for USD 60 million. The announcement period abnormal returns—over 1 day (CAR), over 10 days (CAR10), over 3 days (CAR3)—are all positive, at 2.42%, 3.16%, 1.61%, respectively. The former CEO of Flir, said that the acquisition would reduce the cost of thermal imagining technologies, and would expand product range and distribution channels for Flir. However, from annual reports, net income decreased after the acquisition, and the cost of goods sold increased to USD 697 m, USD 724 m and USD 755 m, respectively, in the 3 years post-acquisition. The ratio of the net income divided the cost of goods sold decreased to 0.25 and 0.27, respectively, in the 2 years post-acquisition, but the ratio was about 0.36 of the year when the acquisition happened and 0.51 three years before the acquisition. Hence, the acquisition seems to have not improved the profitability of the Flir, but instead lowered the operating efficiency post acquisition. In 2018, Flir sold Lorex along with its Toronto-headquartered small and medium-sized security products business for about $23.6 million, much lower than its initial acquisition price. Thus, the announcement period abnormal return seems a misreaction.

In another example, in 2011, Finisar Corp completed acquisition of the entire equity interest in Ignis ASA, which is a provider of optical components and network solutions for

fiber optic communications for NOK 8 per share, the aggregate price is approximately USD 76 million. The offer price represented a premium of 58.4% over the closing price of Ignis on 21 March 2011, the last trading day prior to Finisar's announcement. Considering the high premium, the market reacted negatively to the acquisition, the announcement period abnormal returns—CAR, CAR10, CAR3—are all negative, at −5%, −52%, and −1.57%, respectively. However, this vertical integration decreased the operating expense of Finisar from USD 7.46 m (the year when the acquisition happened) to USD 5.56m (three years after the acquisition). Net income increased from USD 88.10 m (the year when the acquisition happened) to USD 111.79 m (three years after the acquisition). The ratio of the net income divided the cost of goods sold increased from 0.15 (the year when the acquisition happened) to 0.16 (three years after the acquisition)—the profitability of Finisar improved. Through the acquisition, Finisar attained access to an internal source of tunable lases or use of these products. Because of this, Finisar was willing to pay a higher acquisition premium, to which the market may have reacted negatively upon announcement. So, again, the announcement period abnormal return seems a misreaction.

*4.5. Additional Checks*

4.5.1. PSM Method

We check our results using the (propensity score match) PSM method. The main advantage of the *PSM-DID* method is that we can avoid any uncontrolled-for factors in the common trend test and could enhance comparability of positive CAR and negative CAR subsamples. We select the kernel-matching *PSM-DID* to analyze the sample in this robustness check. Propensity score methods (PSM) could minimize selection bias in the non-experimental studies (Rosenbaum and Rubin 1983), could lead to more robust inferences by reducing extrapolation and subsequent dependence on the outcome model specification, and make balancing approach more feasible by condensing covariates into a scalar summary. In the *PSM-DID* model, we select six covariates: Acquirer's Size (*Size*), Leverage (*Lev*), Profit (*Pro*), Income (*Inc*), Cost (*Cost*), and Value of transaction (*VOT*). A non-value-creating acquisition may have different effects on acquirer operating efficiency based on acquirer size and transaction size. Higher leverage could affect profitability and operating efficiency. Profit, income, and cost affect acquirer's profitability, which is pivotal for acquirer's operating efficiency. We do not include the target nation characteristics, as they do not determine the operating efficiency before the acquisition. From Table 8A, the coefficient of the diff-in-diff item is not significant, which corroborates the result we get in the Model (1). In panel B, we could see that all covariates are significant and in accordance with the result we obtained in the previous models. Therefore, we conclude that the association between the announcement period acquirer CAR and merger efficiency is not significant. Note that almost all coefficients are close to zero, which corroborates the finding that a difference in CAR is not associated with change in merger efficiency.

Table 8 shows the result of the PSM-DID model. Panel A reports the difference-in difference estimation result. Panel B reports the regression result of the PSM-DID model. The covariates in the PSM-DID model consists of six variables: Acquirer's Size (Size), Leverage (Lev), Profit (Pro), Income (Inc), Cost (Cost), and Value of transaction (VOT). All variables are defined in Appendix A.

**Table 8.** Robustness check using the PSM-DID model.

| | Panel A | |
| --- | --- | --- |
| | **Coefficient** | ***p*-Value** |
| Diff-in-Diff | −0.000 | 0.843 |
| | **Panel B** | |
| **Variable Name** | **Coefficient** | ***p*-Value** |
| Size | −0.0317 | 0.161 |
| Leverage | −0.0541 | 0.075 * |
| Profit | 0.0607 | 0.472 |
| Cost | −0.0303 | 0.824 |
| Expense | −0.0118 | 0.271 |
| Income | −0.0277 | 0.707 |
| Value of Transaction | 0.5526 | 0.043 ** |

* denotes significant at 10% level, ** significant at 5% level.

### 4.5.2. Alternative Announcement Period Abnormal Returns

We also define CAR in four other ways as a robustness check—CAR10, CAR3, CAR_EW1, and CAR_VW1. CAR10 is the acquirer's stock return minus beta times the S&P 500 return in the ten days around announcement date, CAR3 is the acquirer's stock return minus beta times the S&P 500 return in the three days around announcement date, CAR_EW1 is the acquirer's stock return minus beta times the equally weighted-CRSP-index return in one day around announcement date, and CAR_VW1 is the acquirer's stock return minus beta times the VW CRSP return in one day around announcement date. We re-applied the DID model and the PSM-DID model to obtain new results that were previously obtained in Tables 6 and 8. Table 9 shows that our main results do not change.

Table 9 is Robustness checks with different definitions of CAR. The first 2 panels of Table 9 show regression coefficients, *p*-values in parenthesis and the adjusted $R^2$. The premise that we could employ the DID model is that the sample meets the common trend assumption. Hence, we report the result of the Model (2) in the first panel and the results of Model (1) in the second panel. The next 2 panels report the result of the PSM-DID model. These 4 panels are reported for different alternative methods of computing CAR10, CAR3, CAR-EW1, and CAR-VW1, in sequence.

**Table 9.** Robustness checks with different definitions of CAR.

| | CAR10 | | | |
| --- | --- | --- | --- | --- |
| | **Positive CAR $\times$ Year$_{-3}$** | **Positive CAR $\times$ Year$_{-2}$** | **Control Variables** | **Adj-R$^2$** |
| Coefficient | 0.00002 | $7.44 \times 10^{-6}$ | Included | 0.1284 |
| *p*-Value | 0.280 | 0.697 | | |
| | **Positive CAR** | **Announcement Period** | **Positive CAR $\times$ Announcement Period** | **Control Variables** | **Adj-R$^2$** |
| Coefficient | $7.83 \times 10^{-6}$ | −0.00009 | $6.24 \times 10^{-6}$ | Included | 0.19 |
| *p*-Value | 0.908 | 0.817 | 0.822 | | |
| | **Coefficient** | **p-Value** |
| Diff-in-Diff | −0.000 | 0.972 |
| **Variable Name** | **Coefficient** | **p-Value** |
| Size | 0.0229 | 0.286 |
| Leverage | −0.0032 | 0.915 |
| Profit | 0.0377 | 0.256 |
| Cost | 0.0454 | 0.241 |
| Expense | 0.0457 | 0.350 |
| Income | 0.0708 | 0.306 |
| Value of Transaction | −0.0751 | 0.543 |

**Table 9.** *Cont.*

| | CAR3 | | | |
|---|---|---|---|---|
| | **Positive CAR × Year$_{-3}$** | **Positive CAR × Year$_{-2}$** | **Control Variables** | **Adj-R$^2$** |
| Coefficient | $-5.51 \times 10^{-6}$ | $-0.00002$ | Included | 0.13 |
| *p*-Value | 0.769 | 0.215 | | |
| | **Positive CAR** | **Announcement Period** | **Positive CAR × Announcement Period** | **Control Variables** | **Adj-R$^2$** |
| Coefficient | 0.00019 | $-0.00008$ | $-0.00002$ | Included | 0.19 |
| *p*-Value | 0.005 *** | 0.830 | 0.563 | | |

| | **Coefficient** | ***p*-Value** |
|---|---|---|
| Diff-in-Diff | $-0.000$ | 0.767 |

| **Variable Name** | **Coefficient** | ***p*-Value** |
|---|---|---|
| Size | $-0.0227$ | 0.286 |
| Leverage | $-0.0559$ | 0.066 * |
| Profit | 0.0082 | 0.706 |
| Cost | 0.0168 | 0.533 |
| Expense | $-0.0647$ | 0.237 |
| Income | $-0.0225$ | 0.743 |
| Value of Transaction | $-0.0325$ | 0.841 |

| | CAR-EW1 | | | |
|---|---|---|---|---|
| | **Positive CAR × Year$_{-3}$** | **Positive CAR × Year$_{-2}$** | **Control Variables** | **Adj-R$^2$** |
| Coefficient | $-5.51 \times 10^{-6}$ | $-0.00002$ | Included | 0.13 |
| *p*-Value | 0.769 | 0.215 | | |
| | **Positive CAR** | **Announcement Period** | **Positive CAR × Announcement Period** | **Control Variables** | **Adj-R$^2$** |
| Coefficient | 0.00019 | $-0.00008$ | $-0.00002$ | Included | 0.19 |
| *p*-Value | 0.005 *** | 0.830 | 0.563 | | |

| | **Coefficient** | ***p*-Value** |
|---|---|---|
| Diff-in-Diff | $-0.000$ | 0.790 |

| **Variable Name** | **Coefficient** | ***p*-Value** |
|---|---|---|
| Size | $-0.0227$ | 0.286 |
| Leverage | $-0.0559$ | 0.066 * |
| Profit | 0.0082 | 0.706 |
| Cost | 0.0168 | 0.533 |
| Expense | $-0.0647$ | 0.237 |
| Income | $-0.0225$ | 0.743 |
| Value of Transaction | $-0.0325$ | 0.841 |

| | CAR-VW1 | | | |
|---|---|---|---|---|
| | **Positive CAR × Year$_{-3}$** | **Positive CAR × Year$_{-2}$** | **Control Variables** | **Adj-R$^2$** |
| Coefficient | $-5.51 \times 10^{-6}$ | $-0.00002$ | Included | 0.13 |
| *p*-Value | 0.769 | 0.215 | | |
| | **Positive CAR** | **Announcement Period** | **Positive CAR × Announcement Period** | **Control Variables** | **Adj-R$^2$** |
| Coefficient | 0.00019 | $-0.00008$ | $-0.00002$ | Included | 0.19 |
| *p*-Value | 0.005 *** | 0.830 | 0.563 | | |

| | **Coefficient** | ***p*-Value** |
|---|---|---|
| Diff-in-Diff | $-0.000$ | 0.862 |

| **Variable Name** | **Coefficient** | ***p*-Value** |
|---|---|---|
| Size | $-0.0227$ | 0.286 |
| Leverage | $-0.0559$ | 0.066 * |
| Profit | 0.0082 | 0.706 |
| Cost | 0.0168 | 0.533 |
| Expense | $-0.0647$ | 0.237 |
| Income | $-0.0225$ | 0.743 |
| Value of Transaction | $-0.0325$ | 0.841 |

* denotes significant at 10% level, and *** significant at 1% level.

### 4.5.3. Subsample Analysis

In another check, we segregate our sample into M&A's with positive post-merger operating efficiency and negative post-merger operating, to check associations of CAR in each subsample. However, we find that (in Table 10) that coefficients of *Positive CAR* $\times$ *Year$_{-3}$* and *Positive CAR* $\times$ *Year$_{-2}$* are significant, hence do not meet the common trend assumption, which is the premise of the DID model. In the other word, differences in operating efficiency of these sub-samples exist before the merger. Therefore, we cannot conclude that CAR is significantly associated with post-merger operating efficiency in these sub-samples.

Table 10 shows regression coefficients, *p*-values in parenthesis and the adjusted R$^2$. The premise that we could employ the DID model is that the sample meets the common trend assumption. Hence, we report the result of the Model (2) in the panel. In this table, we segregate our sample into M&As with positive post-merger operating efficiency and negative post-merger operating efficiency.

**Table 10.** Robustness check using different samples.

| | Samples with Positive Post-Merger Efficiency | | | |
|---|---|---|---|---|
| | Positive CAR $\times$ Year$_{-3}$ | Positive CAR $\times$ Year$_{-2}$ | Control Variables | Adj-R$^2$ |
| Coefficient | 0.00006 | 0.00006 | Included | 0.16 |
| *p*-Value | 0.005 *** | 0.006 *** | | |
| | Samples with Negative Post-Merger Efficiency | | | |
| | Positive CAR $\times$ Year$_{-3}$ | Positive CAR $\times$ Year$_{-2}$ | Control Variables | Adj-R$^2$ |
| Coefficient | −0.00009 | −0.00013 | Included | 0.11 |
| *p*-Value | 0.019 ** | 0.000 *** | | |

** denotes significant at 5% level, and *** significant at 1% level.

### 5. Conclusions

Using a dataset of 822 cross-border acquisitions conducted by U.S. companies, spanning a 24-year period from 1990 through 2013, we show that cross-border M&As do not improve the acquirer's operating efficiency; instead, they decrease the acquirer's operating efficiency, on average, post-acquisition. Indeed, operating efficiency was significantly higher two and three years before the acquisition announcement, as compared to the year of the announcement. Three years after the acquisition, the operating efficiency decreases significantly further as compared to the transaction year.

We then examine whether the short-term stock market reaction of acquirers to cross-border M&A announcements reflect longer-term post-merger operating efficiency. Our results suggest that the announcement-period acquirer CAR has no significant relationship with post-merger efficiency. Indeed, acquirers that experienced negative CAR experienced a significant decrease in operating efficiency pre-acquisition, while the acquirers with positive CAR did not. Hence, we conclude that the CAR during the acquisition may be a reflection of the acquirer's operating efficiency *before* the acquisition. Therefore, investors should be careful interpreting announcement period stock price returns in cross-border mergers and acquisitions.

There were several ways in which this study can be expanded. There may be other factors that may influence post-acquisition efficiency, which may need to be accounted for. These may include internal factors such as the introduction of new product lines (Dutordoir et al. 2012) and external factors that may include changes in economic conditions (Medovikov 2016). For example, it will be interesting to examine announcement period market reactions and post-acquisition operating efficiency changes in light of the disruptions caused by the COVID-19 pandemic on the economy (see, e.g., Batool et al. 2020). Future work may also delve deeper into the reasons for investor misreaction, which may be

reactionary, for example, based on recent effects such as the economic crisis, or behavioral, such as expectations for different industries, at different points in time.

**Author Contributions:** Conceptualization, C.N.V.K. and J.W.; methodology J.W.; validation, C.N.V.K.; formal analysis, J.W.; investigation, C.N.V.K.; writing—review and editing, C.N.V.K. All authors have read and agreed to the published version of the manuscript.

**Data Availability Statement:** Refinitiv's SDC Platinum Mergers and acquisitions database; Center for Research in Security prices database, and S&P Compustat database.

**Conflicts of Interest:** The authors declare no conflict of interest.

## Appendix A. Definition of Variables

| Variables | Description |
| --- | --- |
| Cumulative Abnormal Return (CAR) | Cumulative abnormal return is defined as return of the acquirer's stock minus beta times S&P return, where beta is estimated by regressing over the year prior to the announcement. Computed over 1 day, 10 days (CAR10), 3 days (CAR3) around the merger and acquisition announcement date, and over and above the equally weighted CRSP return (CAR_EW1), or value-weighted CRSP return (CAR_VW1), using data from CRSP |
| Total Assets | Total Assets of Acquirer, taken from annual Compustat |
| Total Cost | Total Cost of Acquirer, includes prime operating coat and tax, taken from annual Compustat |
| Total Operating Expense | Total Operating Expense of Acquirer, includes the management expense, financial expense and sales expense, taken from annual Compustat |
| Net Income | Net Income of Acquirer, taken from annual Compustat |
| Prime Operating Revenue | Prime Operating Revenue of Acquirer, taken from annual Compustat |
| Total Debt | Total Debt of Acquirer, taken from annual Compustat |
| Size | Computed as Ln (Total Assets) |
| Leverage | Computed as (Total Debt)/(Total Assets) |
| Profit | Computed as (Net income)/(Prime Operating Revenue) |
| Income | Computed as (Prime Operating Revenue)/(Total Assets) |
| Cost | Computed as (Total Cost)/(Prime Operating Revenue) |
| Expense | Computed as (Management Expense + Financial Expense + Sales expense)/(Prime Operating Revenue) |
| Control and Corruption | Perceptions of the extent to which public power is exercised for private gain, including both petty and grand forms of corruption, as well as "capture" of the state by elites and private interests<br>(Source: Global Insight Business Conditions and Risk Indictors) |
| Government Effectiveness | Perceptions of the quality of public services, the quality of the civil service and the degree of its independence from political pressures, the quality of policy formulation and implementation, and the credibility of the government's commitment to such policies.<br>(Source: Global Insight Business Conditions and Risk Indictors) |
| Rule and Law | Perceptions of the extent to which agents have confidence in and abide by the rules of society, and in particular the quality of contract enforcement, property rights, the police, and the courts, as well as the likelihood of crime and violence<br>(Source: Global Insight Business Conditions and Risk Indictors) |
| Regulatory Quality | Perceptions of the ability of the government to formulate and implement sound policies and regulations that permit and promote private sector development<br>(Source: Global Insight Business Conditions and Risk Indictors) |

| Variables | Description |
|---|---|
| Political Stability and Absence of Violence/Terrorism | Perceptions of the likelihood of political instability and/or politically motivated violence, including terrorism (Source: Global Insight Business Conditions and Risk Indictors) |
| Voice and Accountability | Perceptions of the extent to which a country's citizens are able to participate in selecting their government, as well as freedom of expression, freedom of association, and a free media (Source: Global Insight Business Conditions and Risk Indictors) |
| Rates of Adherence per 1000 Population | The religious adherence per 1000 people of the county in which the acquirer is located, as reported in *Association of Religion Data Archives* |
| Value of Transaction | Value of Transaction/Total Assets of the acquirer |
| External Investment | Total External Investment/Total GDP of target nation |
| GDP Growth, Inflation Rate, Tax Rate | All the target nation |

## Note

[1] See, for example, https://www.publichealth.columbia.edu/research/population-health-methods/difference-difference-estimation assessed on 9 August 2018.

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
