# Peer review of "Market Misreaction? Evidence from Cross-Border Acquisitions"

_jrfm, doi:10.3390/jrfm15020093_

Round 1
Reviewer 1 Report
The study addresses an interesting topic.
But, the article in its present form does not meet the requirements of scientific research, the paper must be reorganized and improved. The paper must be completely restructured and each section clearly presented:
Abstract:
- must contain a summary of the study conducted by the authors, presenting the objectives pursued, its methodology, and conclusions
Introduction:
- The authors should define clearly the purpose of the paper the main objectives of the study, structure of the paper, methodology used and finally they must mention the main conclusions. The authors could formulate some research hypotheses to achieve the set objective, which are then to verify and establish the conclusions in the paper;
Literature review
- The literature review section must be restructured. The authors must introduce in the ”Introduction” what the authors have proposed and achieved in the paper. In this section, just have must they review the literature on the objectives of the research and the used methodology.
Methodology
- The third section can be called "Data and methodology", and restructured to highlight the models used in the analysis (Bootstrap-DEA model, Difference-in Difference model).
Conclusion
- What the authors present in the conclusions, the analysis of the models used would be more appropriate to present in a section of results and discussions.
- The authors must draw more conclusions which to be sustained by evidence from their research;
- At the same time, the final section of the paper could discuss the most important policy implications emerging from the study and some discussion about the potential generalization of the results obtained by the authors in this study.
The authors should re-examine their manuscript to improve it since the English need to be improved in some places.
Author Response
Reviewer 1
Thank you for all your suggestions and comments. They have been addressed as below:
Abstract:
- must contain a summary of the study conducted by the authors, presenting the objectives pursued, its methodology, and conclusions
Done. Please see the modified abstract.
Introduction:
- The authors should define clearly the purpose of the paper the main objectives of the study, structure of the paper, methodology used and finally they must mention the main conclusions. The authors could formulate some research hypotheses to achieve the set objective, which are then to verify and establish the conclusions in the paper;
Done. Please see the modified introduction.
Literature review
- The literature review section must be restructured. The authors must introduce in the ”Introduction” what the authors have proposed and achieved in the paper. In this section, just have must they review the literature on the objectives of the research and the used methodology.
Done. Please see the modified Literature review.
Methodology
- The third section can be called "Data and methodology", and restructured to highlight the models used in the analysis (Bootstrap-DEA model, Difference-in Difference model).
Done. Please see new section 3.1
Conclusion
- What the authors present in the conclusions, the analysis of the models used would be more appropriate to present in a section of results and discussions.
- The authors must draw more conclusions which to be sustained by evidence from their research;
- At the same time, the final section of the paper could discuss the most important policy implications emerging from the study and some discussion about the potential generalization of the results obtained by the authors in this study.
Done. Please see new section 4.3 Results and Discussion and the modified Conclusion, which includes suggestion made by another reviewer.
Reviewer 2 Report
The authors should consider the following recommendations in order to improve the original manuscript:
- To include certain relevant research questions
- To include the structure of the paper in the Introduction section.
- To extend the Theoretical Framework, by providing more relevant literature review, especially studies conducted during the last 5 years. I suggest extending the literature section by including more recent and relevant studies.
- Why the sample period 1990- 2013? Why not a more current time frame which can also covers the Covid-19 pandemic considering that we are now in 2022 (so there is a significant gap between 2013 and 2022 !!!). Authors should take into consideration much more recent publications in the sphere of discussed subject matter, especially studies conducted during the last 5 years. Please discuss about Covid-19 pandemic caused by Severe Acute Respiratory Syndrome Coronavirus 2 (SARS-CoV-2) and its impact on economy. You can not ignore this global health crisis. I suggest extending the literature section by including recent and relevant studies, such as for instance:
- Batool, M., Ghulam, H., Hayat, M.A., Naeem, M.Z., Ejaz, A., Imran, Z.A., Spulbar, C., Birau, R. & Gorun, T.H. (2020) How COVID-19 has shaken the sharing economy? An analysis using Google trends data, Economic Research-Ekonomska Istraživanja, DOI: 10.1080/1331677X.2020.1863830;
- Deepen the description of the limitations of conducted research and indicate the trends for further empirical research.
- To expand the managerial implications in the article.
- The sources must be added under each table and figure.
- The Conclusions section is a total chaos with all those tables included without any logic, without any justification or suitable reason for this unfortunate choice of authors.
- Tables and figures are framed incorrectly in the paper and do not follow JRFM standards for authors.
- Human proofreading, English grammar and spelling correction are also required in order to improve the quality of the manuscript.
Author Response
Reviewer 2
Thank you for all your suggestions and comments. They have been addressed as below:
- To include certain relevant research questions
Done. Please see modified Introduction
- To include the structure of the paper in the Introduction section.
Done. Please see modified Introduction
- To extend the Theoretical Framework, by providing more relevant literature review, especially studies conducted during the last 5 years. I suggest extending the literature section by including more recent and relevant studies.
Done. Also following specific suggestions of another reviewer.
- Why the sample period 1990- 2013? Why not a more current time frame which can also covers the Covid-19 pandemic considering that we are now in 2022 (so there is a significant gap between 2013 and 2022 !!!). Authors should take into consideration much more recent publications in the sphere of discussed subject matter, especially studies conducted during the last 5 years.
Unfortunately, this will not be possible, as the data comes from a database for which we needed to pay, and which we don’t have access to, anymore. We have analyzed a recent almost quarter century, allowing some time for post-acquisition-period analysis also. So the findings, in terms of investor behavior, are over a long period of time and relatively current.
- Please discuss about Covid-19 pandemic caused by Severe Acute Respiratory Syndrome Coronavirus 2 (SARS-CoV-2) and its impact on economy. You can not ignore this global health crisis. I suggest extending the literature section by including recent and relevant studies, such as for instance:
- Batool, M., Ghulam, H., Hayat, M.A., Naeem, M.Z., Ejaz, A., Imran, Z.A., Spulbar, C., Birau, R. & Gorun, T.H. (2020) How COVID-19 has shaken the sharing economy? An analysis using Google trends data, Economic Research-Ekonomska Istraživanja, DOI: 10.1080/1331677X.2020.1863830;
Deepen the description of the limitations of conducted research and indicate the trends for further empirical research.
To expand the managerial implications in the article.
Done, we have added the references and also added implications in the modified conclusion (that another reviewer has also suggested).
- The sources must be added under each table and figure.
Done.
- The Conclusions section is a total chaos with all those tables included without any logic, without any justification or suitable reason for this unfortunate choice of authors.
- Tables and figures are framed incorrectly in the paper and do not follow JRFM standards for authors.
- Human proofreading, English grammar and spelling correction are also required in order to improve the quality of the manuscript.
All done, and will continue to be done with the editor
Reviewer 3 Report
The authors analyse the misreaction of cross-border acquisitions. They propose the bootstrap-dea model to measure operational efficiency.
Some suggestions and questions about this paper are proposed below.
- From a formal point of view, authors should review some formulas, such as those in Section 4.3. They should be centred and numbered to the right of them. In addition, the tables in the appendix definition of variables on page 16 onwards should also be centred.
- In section 2, other articles could be cited that propose another type of DEA model such as the inverse DEA model. For example:
Guijarro, F., Martínez-Gómez, M., & Visbal-Cadavid, D. (2020). A model for sector restructuring through genetic algorithm and inverse DEA. Expert Systems with Applications, 154, 113422. https://doi.org/10.1016/j.eswa.2020.113422
Or for example in:
Krishnan, C. N. V. and Yakimenko, Vasiliy, Market Misreaction? Leverage and Mergers and Acquisitions (January 22, 2022). Available at SSRN: https://ssrn.com/abstract=3994811 or http://dx.doi.org/10.2139/ssrn.3994811
- Perhaps, in the section 3 should justify the variables chosen for de model
Author Response
Reviewer 3
Thank you for all your suggestions and comments. They have been addressed as below:
- From a formal point of view, authors should review some formulas, such as those in Section 4.3. They should be centred and numbered to the right of them. In addition, the tables in the appendix definition of variables on page 16 onwards should also be centred.
Done
- In section 2, other articles could be cited that propose another type of DEA model such as the inverse DEA model. For example:
Guijarro, F., Martínez-Gómez, M., & Visbal-Cadavid, D. (2020). A model for sector restructuring through genetic algorithm and inverse DEA. Expert Systems with Applications, 154, 113422. https://doi.org/10.1016/j.eswa.2020.113422
Or for example in:
Krishnan, C. N. V. and Yakimenko, Vasiliy, Market Misreaction? Leverage and Mergers and Acquisitions (January 22, 2022). Available at SSRN: https://ssrn.com/abstract=3994811 or http://dx.doi.org/10.2139/ssrn.3994811
Added
- Perhaps, in the section 3 should justify the variables chosen for de model
Done
Round 2
Reviewer 1 Report
In this version of the updated paper, the authors took into account my observations, and thus I consider as their study improved ...
But I have a suggestion looking at the tables at the end of the Conclusions section must either be integrated into the text, where they are referred to and explain what they contain or are listed as Appendices and in this case, it is not necessary to say explication what they contain.
Author Response
In this version of the updated paper, the authors took into account my observations, and thus I consider as their study improved ...
But I have a suggestion looking at the tables at the end of the Conclusions section must either be integrated into the text, where they are referred to and explain what they contain or are listed as Appendices and in this case, it is not necessary to say explication what they contain.
The tables are now integrated into the text.
Highlighted.
Thanks for all your comments and suggestions.
Reviewer 2 Report
The article in the initial version has been improved based on the previous review report.
Author Response
The article in the initial version has been improved based on the previous review report.
Thanks for all your comments and suggestions.